# Intestinal Microbiota of Grass Carp Fed Faba Beans: A Comparative Study

**DOI:** 10.3390/microorganisms7100465

**Published:** 2019-10-17

**Authors:** Lei Zhou, Ke-tao Lin, Lian Gan, Ji-jia Sun, Chang-jun Guo, Li Liu, Xian-de Huang

**Affiliations:** 1Joint Laboratory of Guangdong Province and Hong Kong Region on Marine Bioresource Conservation and Exploitation, College of Marine Sciences, South China Agricultural University, Guangzhou 510642, China; zhoulei@scau.edu.cn (L.Z.); ganlian@scau.edu.cn (L.G.); jjsun@scau.edu.cn (J.-j.S.); 2Institute of Aquatic Economic Animals and Guangdong Province Key Laboratory for Aquatic Economic Animals, School of Life Sciences, Sun Yat-sen University, Guangzhou 510275, China; gchangj@mail.sysu.edu.cn

**Keywords:** grass carp, faba beans, 16S sequencing, intestinal microbiota, *Acinetobacter*

## Abstract

Many reports of the intestinal microbiota of grass carp have addressed the microbial response to diet or starvation or the effect of microbes on metabolism; however, the intestinal microbiota of crisp grass carp has yet to be elucidated. Moreover, the specific bacteria that play a role in the crispiness of grass carp fed faba beans have not been elucidated. In the present study, 16S sequencing was carried out to compare the intestinal microbiota in the fore-, mid- and hind-intestine segments of grass carp following feeding with either faba beans or formula feed. Our results showed that (1) the hind-intestine presented significant differences in diversity relative to the fore- or midintestine and (2) faba beans significantly increased the diversity of intestinal microbiota, changed the intestinal microbiota structure (Fusobacteria was reduced from 64.26% to 18.24%, while Proteobacteria was significantly increased from 17.75% to 51.99%), and decreased the metabolism of energy, cofactors and vitamins in grass carp. Furthermore, at the genus and species levels, *Acinetobacter* accounted for 15.09% of the microbiota, and *Acinetobacter johnsonii* and *Acinetobacter radioresistens* constituted 3.41% and 2.99%, respectively, which indicated that *Acinetobacter* of the family Moraxellaceae contributed to changes in the intestinal microbiota structure and could be used as a potential biomarker. These results may provide clues at the intestinal microbiota level to understanding the mechanism underlying the crispiness of grass carp fed faba beans.

## 1. Introduction

An accidental discovery was made in the early 1970s in Dongsheng Town, Guangdong Province, China, that feeding crisp grass carp (*Ctenopharyngodon idellus* C.et V) that had reached a certain level of maturity faba beans led to carp that exhibited more hardness and crispiness in its muscles than ordinary grass carp, and this diet has since become popular. The crisp mechanism of grass carp may be due to a certain factor in faba beans that activates the TGF-β/Smads signaling pathway to regulate type I collagen in muscles, which is the main type of collagen and is closely related to changes in muscle properties [1,2,3]. Furthermore, a proteomic profile demonstrated that muscle fiber hyperplasia was closely related to a protein-protein network of 12 muscle component proteins, and the abundance of the fatty acid degradation and calcium signaling pathways was reduced. In addition, the pentose phosphate pathway-induced metabolism of grass carp was downgraded, which led to hemolysis after grass carp were fed only whole faba beans [4]. In addition, many studies also focused on various associated subjects, such as the muscle nutrient composition and microstructure, growth performance, blood physiological and biochemical characteristics involved in the crisp process [3,5,6,7]. These studies helped reveal the mechanism of muscle crispiness in grass carp.

The intestinal microbiota participates in many important physiological processes of the host, such as growth, digestion, mucosal immunity system development, and protection against disease, and has integrated and coevolved with the host [8,9,10]. There are many reports on the intestinal microbiota of grass carp from the perspectives of the composition and diversity of the bacterial community [5,11,12,13]. Other studies have focused on responses of the microbiota to diet or starvation or its effect on metabolism, and different intestinal segments have different microbiota that play different roles [14,15,16,17]. However, only a few studies have focused on the intestinal microbiota of crisp grass carp [18]. The intestinal microbiota and the specific bacteria that play a role in the crispiness of grass carp fed faba beans remain unclear.

In this study, we set up a feeding trial for grass carp in which faba beans were used for the treatment group and formula feed was used for the control group, and we investigated the corresponding intestinal microbiota by using Illumina-based high-throughput 16S sequencing. The microbial profiles of the fore-, mid- and hind-intestine segments in both groups were compared, and different microbiota were analyzed to explore the causative mechanism of the potential crispiness of grass carp at the intestinal microbiota level.

## 2. Materials and Methods

### 2.1. Fish and Sample Collection

A total of 50 grass carp with an initial mean weight of (546.3 ± 32.04 g) were randomly divided into two groups (treatment and control group) and cultured in two ponds (6 × 2 × 1.2 m). The control (C) group was fed a formulated diet (Tongwei, China), and the treatment (T) group was fed only faba beans. Fish were fed twice daily at 8:30 and 17:00 for 90 days, and the feed rate was adjusted each month to ensure that they consumed an amount equivalent to approximately 1% of their body weight per day. Six grass carp were randomly collected from each group and the average weights were approximately 1.3 kg for the C group and 0.8 kg for the T group. The entire intestinal tracts were obtained, and the intestinal microbial samples from the fore-, mid- and hind-intestine segments (F, M and H, respectively) were collected by careful scraping with a sterile spatula 2.0 mL cryotube. Each sample was immediately frozen in liquid nitrogen and then transferred to a −80 °C refrigerator until DNA extraction. All animal-involving experiments of this study were approved by the Animal Care and Use Committee of South China Agricultural University (establishment date: 25 March 2014) with the permission No. 20190136 and all efforts were made to minimize suffering.

### 2.2. DNA Extraction, PCR Amplification and Illumina HiSeq 2500 Sequencing

Microbial DNA was extracted from these samples by using the E.Z.N.A. stool DNA Kit (Omega Biotek, Georgia, USA) according to the manufacturer’s protocols. The 16S rDNA V3-V4 region of the Eukaryotic ribosomal RNA gene was amplified by PCR using the primers 341F: CCTACGGGNGGCWGCAG and 806R: GGACTACHVGGGTATCTAAT (where the barcode is an eight-base sequence unique to each sample). PCR assays were performed at 95 °C for 2 min; followed by 27 cycles at 98 °C for 10 s, 62 °C for 30 s, and 68 °C for 30 s; and a final extension at 68 °C for 10 min.

Amplicons were extracted from 2% agarose gels, purified using the AxyPrep DNA Gel Extraction Kit (Axygen Biosciences, California, USA) according to the manufacturer’s instructions and quantified using QuantiFluor-ST (Promega, Wisconsin, USA). Purified amplicons were pooled in equimolar amounts, and paired-end sequences (2 × 250) were obtained with an Illumina HiSeq 2500 system (Illumina, California, USA) at Gene Denovo Biological Technology Co. Ltd. (Guangzhou, China).

### 2.3. Quality Control and Read Assembly

Raw reads were filtered to obtain high-quality clean reads, which were merged as raw tags using FLASH [19] (v 1.2.11), with a minimum overlap of 10 bp and mismatch error rates of 2%. Then, raw tags were filtered by the QIIME (V1.9.1) pipeline under specific filtering conditions to obtain high-quality clean tags [20,21]. Clean tags were searched against the reference database to perform reference-based chimera checking. All chimeric tags were removed, and effective tags were finally obtained for further analysis. Raw read data have been submitted to the NCBI Sequence Read Archive (SRA) under the accession number PRJNA563528.

### 2.4. OTU Cluster and Taxonomy Classification

The effective tags were clustered into operational taxonomic units (OTUs) of ≥ 97% similarity using the UPARSE pipeline [22]. The tag sequence with the highest abundance was selected as the representative sequence within each cluster, and a Venn analysis was performed in R to identify unique and common OTUs.

The representative sequences were classified into organisms by a naive Bayesian model using an the Ribosomal Database Project classifier (Version 2.2) based on the SILVA (v128) database [23] (https://www.arb-silva.de/). 

### 2.5. Diversity Analysis and Functional Prediction

The Chao1, Simpson and other alpha diversity indexes were calculated in QIIME (V1.9.1) as mentioned above. An OTU rarefaction curve and rank abundance curves were plotted in QIIME. Statistics of the alpha index comparison between groups were calculated by Welch’s t-test and the Wilcoxon rank test in R. The alpha index comparison among groups was computed by Tukey’s HSD test and the Kruskal-Wallis H test in R (version 3.5.1).

Nonmetric multidimensional scaling (NMDS) of Bray–Curtis dissimilarity index was calculated and plotted in R. An analysis of similarities (ANOSIM) based on the Bray-Curtis dissimilarity index was performed to determine significant differences in the microbial community. *R* values in ANOSIM were used to detect the overlap in the community as reported by Buttigieg and Ramette (*R* > 0.75, well separated; 0.50 < *R* ≤ 0.75, separated but overlapped; 0.25 < *R* ≤ 0.50, separated but strongly overlapped; and 0.25 ≤ *R*, barely separated) [24]. The *p* value was used to indicate significant differences between the two groups (* *p* < 0.05 and ** *p* < 0.01).

Differential species analyses between the C and T groups were compared and identified using Welch’s t-test in R. Furthermore, to analyze the functions of the intestinal microbiota communities in the C and T groups, the functional prediction of the OTUs was inferred using Tax4Fun (v1.0) [25]. A related analysis was carried out using the online platform of Gene Denovo Biological Technology Co. Ltd. (Guangzhou, China) (http://www.omicsmart.com/).

## 3. Results

### 3.1. Statistics of Illumina Sequencing Data and OTUs

A total of approximately 4.4 M million raw reads of bacterial 16S rDNA were obtained by Illumina sequencing. After data quality filter processing, approximately 4.2 M effective reads/tags were obtained. In detail, the total numbers of OTUs were 1290, 1146, 964 for the TF, TM, and TH treatments, respectively, and 572, 618, 482 for the CF, CM, and CH treatments, respectively. A summary of the OTUs and tags of all samples is listed in Appendix A. 

The differences in OTUs between different samples or groups were illustrated using Venn diagrams, which showed that the total number of OTUs was 1519 for T and 609 for C (Appendix A). In addition, 246 OTUs overlapped for the FI, MI, and HI OTUs in the treatment and control group (Appendix A). Compared with the C group, the T group had more abundant intestinal microbes, suggesting that more intestinal microbes were present to digest, absorb and manage faba beans. Moreover, the number of OTUs from the hind-intestine was significantly less than that from the fore- and mid-intestine for the T and C groups (Appendix A), thereby indicating that the abundance of the microbial population in the hind-intestine was significantly decreased compared with that of the fore- and mid-intestine areas.

### 3.2. Microbiota Structure 

To visualize the variation in species abundance of different samples at different taxonomic levels, taxonomy stack distributions were carried out to statistically analyze the species composition of each sample at each level of classification. In this classification, the top 10 species with an abundance of 2% in the sample are shown, and the rest were unified to the other category. The tags that could not be annotated to a specific taxonomy were classified to the unclassified category (Figure 1 and Appendix A). A summary of the profiling for each sample at all taxonomic levels is listed in Appendix A. 

For the taxonomy stack distributions at the phylum level, the bacterial taxonomic compositions of the CF, CM and CH treatments in the control group showed a relatively high average abundance of Fusobacteria (72.5 ± 6.9%, 74.9 ± 7.8% and 46.4 ± 21%), followed by Proteobacteria (20.7 ± 8.2%, 18.9 ± 3.9%, and 13.6 ± 5.7%), Bacteroidetes (1.6 ± 2%, 3.4 ± 6.9%, and 19.5 ± 14.6%) and Firmicutes (4.2 ± 2.9%, 2.8 ± 1.2%, and 10.4 ± 9.3%), whereas the composition in the TF, TM and TH treatments in the treatment group showed an almost complete relative dominance of Proteobacteria (58.4 ± 12.9%, 55 ± 9.6%, and 42.5 ± 25.4%), followed by Fusobacteria (16.9 ± 10.1%, 20.4 ± 8.8%, and 17.4 ± 12.5%), Bacteroidetes (5.3 ± 3%, 5.9 ± 3.8%, and 21.6 ± 17.3%) and Firmicutes (9 ± 5.8%, 7 ± 3.4%, and 4.9 ± 3.4%). Similar ratios of the bacterial taxonomic composition were observed at the class and order levels, except for the other category, in which the proportion increased in the treatment group. At the family level, the ratios of the other and unclassified categories rapidly increased and even covered two-thirds in the control group. At the species level, the CF, CM and CH of the control group were almost covered by the unclassified category, while the TF, TM and TH of the treatment group showed distributions of approximately 3% for *Acinetobacter johnsonii*, *Acinetobacter radioresistens* and the other category. 

### 3.3. Diversity Differences and Potential Functions

To analyze the microbial community diversity within the sample, alpha diversity (a single sample diversity analysis) was used to reflect the diversity of microbial communities. The four indexes of alpha diversity analysis, i.e., the Chao1, observed species, Shannon, and Simpson indexes, were calculated for different samples using the Wilcoxon rank sum test. Significant differences are listed in Figure 2. A comparison of the T to C group showed significant differences (*p* < 0.05) for the above four indexes (Figure 2A). However, the alpha diversity differences within the group were not significant except for the observed species index for TF/TH, CF/CH and CM/CH (*p* < 0.05) and Shannon index for TF/TH (*p* < 0.05) (Figure 2B). These results indicated that there were significant diversity differences between groups but few differences within groups and that the differences were primarily associated with the differences between the hind-intestine and the fore- or mid-intestine. 

To compare the diversity between different ecosystems and indicate the response of biological species to the environment, NMDS was carried out to analyze the beta diversity based on the Bray–Curtis dissimilarity index of samples. As shown in Figure 3, the NMDS showed that the T and C groups could be closely clustered together, indicating that the similarity among the groups was higher but the difference between the groups was obvious. Furthermore, to detect significant differences in the community of different groups, a pairwise ANOSIM was performed (Table 1), and the results indicated that the groups were well separated for T/C (ANOSIM−R = 0.87, *p* = 0.001), TF/CF (ANOSIM−R = 1.00, *p* = 0.002) and TM/CM (ANOSIM−R = 0.98, *p* = 0.005), separated but slightly overlapped for TH/CH (ANOSIM-R = 0.62, *p* = 0.003), and separated but strongly overlapped for CF/CH (ANOSIM−R = 0.30, *p* = 0.018), CM/CH (ANOSIM−R = 0.28, *p* = 0.017) and TF/TH (ANOSIM−R = 0.34, *p* = 0.049). These results indicated that there were significant differences in the community between groups and that the microbial community of the hind-intestine was significantly separated but strongly overlapped compared to the microbial community of the fore- or mid-intestine.

Furthermore, the functional profiling of the intestinal microbial communities from all samples was predicted from the 16S rRNA gene amplicon data using Tax4Fun (v1.0). As shown in Figure 4, significantly different results indicated that among the five KEGG pathways (Level 1) of metabolism, environmental information processing, genetic information processing, cellular processes, and human diseases, the main changes in the intestinal microbiota gene functions were focused on metabolism, which contained four of the top six differences in the abundance of KEGG pathways (i.e., carbohydrate metabolism, amino acid metabolism, energy metabolism, and metabolism of cofactors and vitamins) (Level 2). Because higher OTU numbers were observed in the T group, the results showed that “signal transduction”, “energy metabolism” and “metabolism of cofactors and vitamins” were significantly decreased in the T group compared to the C group, while “carbohydrate metabolism”, “amino acid metabolism” and “membrane transport” were increased, which suggested that metabolic changes occurred in the intestinal microbiota of grass carp caused by the feeding of broad beans.

The following results were obtained based on the various indicators of the alpha and beta diversity analysis and combined functional prediction: (1) the hind-intestinal microbiota was less diverse than the fore- and mid-intestinal microbiota, and a significant diversity difference was observed between the hind-intestine and the fore- or mid-intestine; and (2) the intestinal microbiota in the corresponding parts between the T and C groups were significantly different, which indicates that the faba bean diet likely changed the intestinal microbiota and their corresponding metabolism. 

### 3.4. Differences at Different Taxonomic Levels

After categorizing the significant changes in microbiota, the five most abundant terms were obtained at different taxonomic levels. As listed in Figure 5, compared to the C group, the T group showed that at the phylum level, Fusobacteria was reduced significantly from 64.26% to 18.24% and Proteobacteria increased significantly from 17.75% to 51.99%. At the class and order levels, similar results were obtained for Fusobacteria and Proteobacteria (including γ-proteobacteria and α-proteobacteria) at the class level and for Fusobacteriales and Pseudomonadales at the order level. Because Fusobacteriales can only be classified as “unclassified” below the order level, levels below family could not be annotated and had no relevant information in this study. At the family level, Moraxellaceae and Caulobacteraceae showed a sharp rise; at the genus level, *Acinetobacter* and *Brevundimonas* increased significantly; and at the species level, *Acinetobacter johnsonii*, *Acinetobacter radioresistens* and *Rhizobium radiobacter* significantly increased. Detailed data are listed in Appendix A.

Considering that α-proteobacteria accounted for a relatively small proportion (10.67%), γ-proteobacteria (36.39%) was the main dominant microbiota of the T group. Additionally, according to the taxonomic and expression profiling data of the species, the schematic diagram (Figure 6) of the Proteobacteria species taxonomic tree was simplified for the main significant genus/species differences in the intestinal microbiota between the T and C groups. The corresponding ratios were as follows: Proteobacteria (17.75%), γ-Proteobacteria (16.75%), Pseudomonas (0.84%), Moraxella (0.76%), *Acinetobacter* (0.72%), *Acinetobacter johnsonii* (0.18%) and *Acinetobacter radioresistens*(0.15%) for the C group; and Proteobacteria (51.99%), γ-Proteobacteria (36.39%), Pseudomonas (17.53%), Moraxellaceae (16.05%), *Acinetobacter* (15.09%), *Acinetobacter johnsonii* (3.41%) and *Acinetobacter radioresistens* (2.99%) for the T group. At the same time, in γ-Proteobacteria, *Aeromonas* had no significant difference between the T and C groups (14.92% and 11.51%), although *Aeromonas* was also the main dominant bacterial genus. Additionally, in Proteobacteria, the secondary corresponding ratios of change in the Alphaproteobacteria and below level were as follows: Alphaproteobacteria (0.52%), Caulobacteraceae (0.18%), Brevundimonas (0.18%) for the C group; and Alphaproteobacteria (10.67%), Caulobacteraceae (4.55%), Brevundimonas (4.36%) for the T group.

The results indicated that a faba bean diet changed the intestinal microbiota structure, and the main difference was that Fusobacteria was reduced. Meanwhile, Proteobacteria increased significantly, which led to significant decreases in “energy metabolism” and “metabolism of cofactors and vitamins” in the T group. Furthermore, at the genus level, *Acinetobacter* of Moraxellaceae mainly played a major role in the changes in the intestinal microbiota structure with the feeding of faba beans to grass carp.

## 4. Discussion

The crispiness of grass carp is an extremely complex physiological and biochemical process that eventually leads to excessive crispiness, which directly leads to death due to blood circulation disorders. The degree of crispiness and physical changes are consistent with the toxicological dose (faba bean)-effect response [26]. Studies have shown that faba beans significantly inhibit growth, decrease the ratio of liver to body and gradually lead to excessive accumulation of fat in the hepatopancreas and abdominal cavity of grass carp [27,28]. Additionally, faba beans may increase the abundance of gram-negative and flagellated bacteria and then result in intestinal inflammation in fish, including grass carp, and these changes share similar features with inflammatory bowel disease (IBD) [18]. 

The composition of fish intestinal microbiota is susceptible to diet, which reflects and plays an important role in the health of the fish intestine [29]. The dominant bacterial species largely determine the function of the intestinal microbiota community of a fish. Thus, understanding the species composition at different taxonomic levels can effectively interpret the formation, changes and ecological impacts of the community structure. Meanwhile, by interacting with the intestinal microbiota, the metabolism in the intestine or host can change and more effectively adapt to changes stemming from new nutrients/environmental conditions. A previous study showed that intestinal microbiota may shape intestinal immune responses in healthy and diseased states [30].

Many studies have described that Proteobacteria blooms in the intestine reflect an unstable microbial community structure or a state of host disease. Moreover, the natural microbial community of the intestines normally contains only a minor proportion of this phylum and is dominated by Firmicutes or Bacteroidetes [31,32]. In contrast, Fusobacteria, a distinct and understudied phylum of bacteria, is divided into two families: Leptotrichiaceae and Fusobacteriaceae. These less-studied bacteria are gram-negative, non-spore-forming, and usually nonmotile anaerobes with unique metabolic capabilities [33]. In this study, a comparison of the C and T groups showed that Fusobacteria was reduced from 64.26% to 18.24% while Proteobacteria increased from 17.75% to 51.99%, thus suggesting that faba beans resulted in an imbalance and changes to the intestinal microbiota in grass carp.

At the family and lower levels, since members of Fusobacteria are unclassified, the dominant species and/or changes were concentrated in the genus *Acinetobacter*, which belongs to the phylum Proteobacteria, class Gammaproteobacteria, order Pseudomonadales, and family Moraxellaceae. *Acinetobacter* contains gram-negative, nonfermentative, nonmotile, oxidase-negative, catalase-positive, and strictly aerobic bacteria with a G + C content of 39–47%, and its species are widely distributed in nature, can survive on various surfaces (both moist and dry) and are spread by various means [34,35,36,37]. As significant opportunistic pathogens, they are generally considered to have low virulence and cause a decrease in body resistance and immune function after infection even though they have an astonishing ability to acquire antibiotic resistance genes [38,39]. In addition, these bacteria contribute to the mineralization of aromatic compounds [40,41]. In this study, *Acinetobacter* accounted for 15.09% and *Acinetobacter johnsonii* and *Acinetobacter radioresistens* accounted for 3.41% and 2.99%, respectively. This result showed that *Acinetobacter* played an important role in the intestinal microbiota by combining with gram-negative and flagellated bacteria introduced by faba beans, which resulted in intestinal inflammation [18,42]. Thus, *Acinetobacter* might lead to intestinal inflammation and furthermore become a potential cause of crispiness in grass carp. Therefore, *Acinetobacter* can also be used as a primary indicator/biomarker for detecting crispiness in grass carp. However, as a fish pathogen, *Acinetobacter* has been rarely reported. Why does *Acinetobacter* but not other Pseudomonas genera, such as *Aeromonas*, dominate the intestinal microbiota of grass carp after faba bean feeding? Is it related to gram-negative bacteria that do not ferment sugars and cannot fully utilize and absorb intestinal decomposition products? To answer these questions, further experiments, such as multiomic combinations, are required. 

In addition, because the intestine is the main site for nutrient absorption, when the bacteria in the intestine are significantly increased, they often cause gastrointestinal symptoms, such as bloating, abdominal pain and diarrhea, affect the absorption of carbohydrates and fats, especially the absorption of fat-soluble vitamins, and worsen intestinal inflammation [43]. Moreover, a high amount of bacteria leads to the competitive absorption of nutrients from the host [44,45]. This study indicates that faba beans significantly increased the intestinal bacteria and decreased the energy metabolism (energy metabolism) and cofactor and vitamin metabolism (Metabolism of Cofactors and Vitamins) in grass carp, which also aggravated the severity of the symptoms. The distribution of microbiota varies in different intestinal segments, and the results indicated that hind-intestinal microbiota were significantly different from the fore- and mid-intestinal microbiota, which is consistent with previous studies [14,15].

In conclusion, 16S sequencing and comparative analyses using faba beans and formula feed were carried out on the intestinal microbiota in different intestine segments of grass carp. Our analysis showed that there were significant differences in diversity between the hind-intestine and the fore- or mid-intestine, and that faba beans shifted the intestinal microbial composition significantly, with the proportion of Fusobacteria decreased, the proportion of *Acinetobacter* increased, and the overall diversity increased. 

## Figures and Tables

**Figure 1 microorganisms-07-00465-f001:**
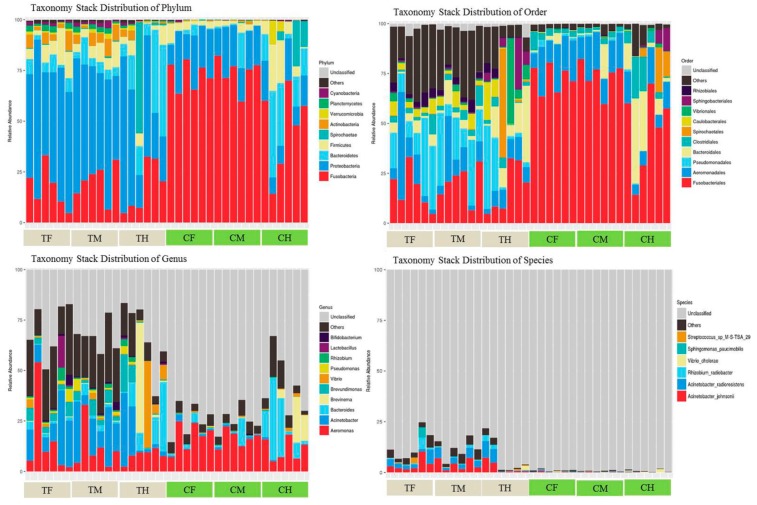
Taxonomy stack distributions for different levels of classification for each group. Relative bacterial abundances at the phylum, order, family and species levels for each group are shown, and the most abundant taxonomic classifications are listed.

**Figure 2 microorganisms-07-00465-f002:**
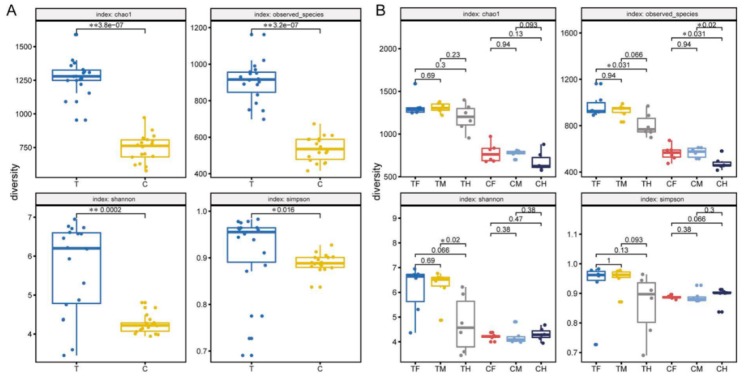
Differences in alpha diversity between groups (**A**) and within groups (**B**). The four indexes of alpha diversity analysis, i.e., the Chao1, observed species, Shannon, and Simpson indexes, were calculated for different samples, and the groups of datawere compared with the Wilcoxon rank sum test.

**Figure 3 microorganisms-07-00465-f003:**
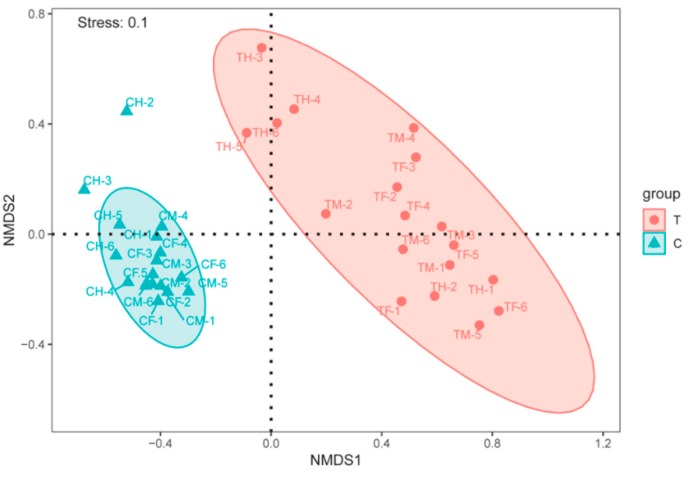
Nonmetric multidimensional scaling (NMDS) was used to analyze the beta diversity based on the Bray–Curtis dissimilarity index of the samples. The samples were divided into two groups.

**Figure 4 microorganisms-07-00465-f004:**
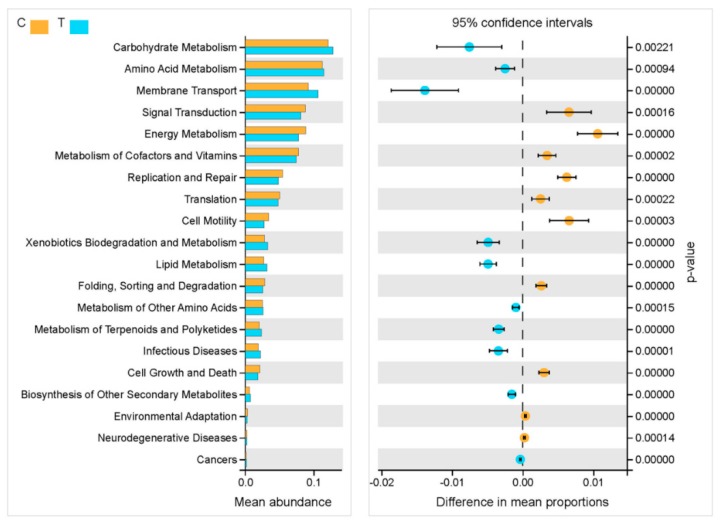
Functional differences in intestinal microbial communities from all samples predicted using Tax4Fun. Significantly different results (*p* < 0.01) among the five KEGG pathway categories are shown.

**Figure 5 microorganisms-07-00465-f005:**
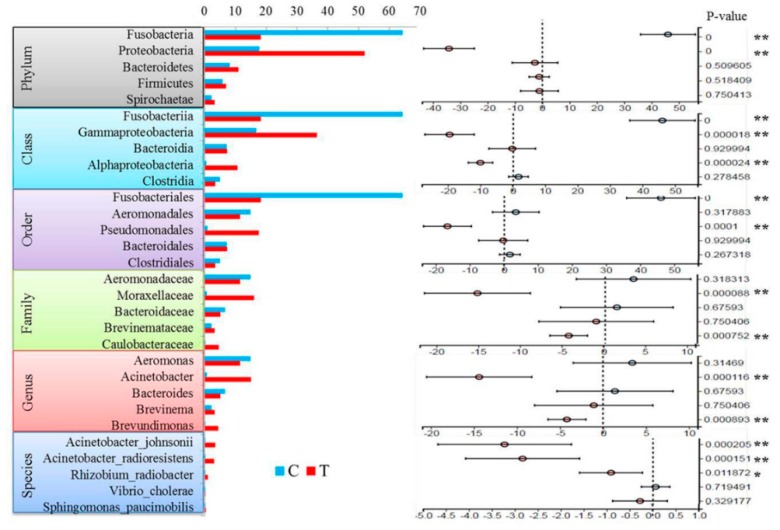
Comparison of the intestinal microbiota abundances in the treatment and control groups at the different taxonomic levels (A). One-way ANOVA bar plot for different taxonomic levels. * *p* < 0.05 and ** *p* < 0.01.

**Figure 6 microorganisms-07-00465-f006:**
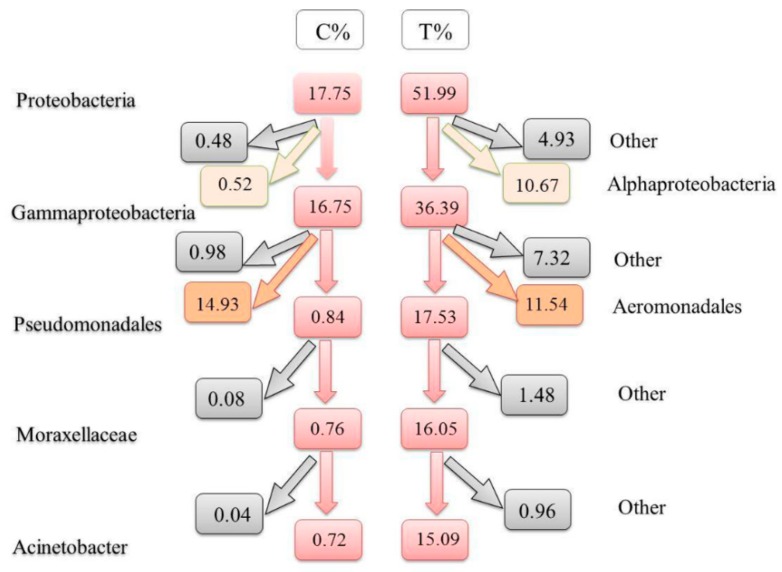
Schematic diagram of the intestinal microbiota abundances at the different taxonomic levels of Proteobacteria in the treatment and control groups.

**Table 1 microorganisms-07-00465-t001:** Pairwise ANOSIM analysis of the different groups.

Group	Distance	R	*p* Value	Significant#
TF/TM	Bray-Curtis	−0.12	0.854	
TF/TH	Bray-Curtis	0.34	0.049	*
TM/TH	Bray-Curtis	0.25	0.073	
CF/CM	Bray-Curtis	−0.04	0.731	
CF/CH	Bray-Curtis	0.30	0.018	*
CM/CH	Bray-Curtis	0.28	0.017	*
TF/CF	Bray-Curtis	1.00	0.002	**
TM/CM	Bray-Curtis	0.98	0.005	**
TH/CH	Bray-Curtis	0.62	0.003	**
T/C	Bray-Curtis	0.87	0.001	**

#: * stands for *p* < 0.05, and ** stands for *p* < 0.01

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
