# Peer review of "Intestinal Microbiota of Grass Carp Fed Faba Beans: A Comparative Study"

_microorganisms, 2019, doi:10.3390/microorganisms7100465_

Round 1

Reviewer 1 Report

Compared to the original manuscript, I think the authors have done a fine job of shoring up some of the statistical comparisons. However the connection between the inflammatory phenotype described in the introduction and the differences in microbial composition described here remains elusive. I think the experiments described here are sound, but this paper may struggle to find an audience. 

Author Response

Response: For the connection between the inflammatory phenotype described in the introduction and the differences in microbial composition described here, "However, only a few studies have focused on the intestinal microbiota of crisp grass carp. Previous studies have shown that faba beans increased the abundance of gram-negative and flagellated bacteria, which resulted in epithelial cell damage and intestinal inflammation in the grass carp [17]. However, the specific bacteria that play a role in the crispiness of grass carp fed faba beans remain unclear." has been changed to "However, only a few studies have focused on the intestinal microbiota of crisp grass carp [17]. The intestinal microbiota and the specific bacteria that play a role in the crispiness of grass carp fed faba beans remain unclear.".

Reviewer 2 Report

Dear editor:

I appreciate the Editor to give me a chance to review this paper. The author reported the intestinal microbiota of grass carp after treatment of faba beans. I suggest the author added the other data of grass carp, such as body weights, in the manuscript. Based on the sequencing of 16S rDNA V3-V4 region, the available information was limited in the present work.

Author Response

Response: The data of body weights have been added in line 68. For the limited available information from sequencing of 16S rDNA, our other works such as transcriptomic sequencing have been performed to increase the available information.

Reviewer 3 Report

Authors significantly corrected manuscript and now I recommend it to publication.

Corrections:

References should be corrected according to Instructions for authors. Figures have poor resolution.

Author Response

Response: References have been corrected according to Instructions for authors. The resolution of the Figure has been already ≥ 300bpi. For example, the resolution of the Figure 1 has increased from 300 bpi to 400 bpi.

Round 2

Reviewer 2 Report

The authors have revised the manuscript. The manuscript could be accepted in the present form.

This manuscript is a resubmission of an earlier submission. The following is a list of the peer review reports and author responses from that submission.

Round 1

Reviewer 1 Report

Reviewed aricle is poor and written weak English. Many sentences are incomprehensible. For Authors is no matter if they write about Pseudomonas, Moraxella or Acinetobacter. Also names of bacterial Class, Order and Family are different, without the right ends. What means "the Acinetobacter spp of Moraxella"? Acinetobacter is not Moraxella! What Authors think writing "Gram-negative and flagellated bacteria". Among Gram-negative bacteria are also flagellated. Introduction should be corrected to be an introduction to this work, because it is currently unknown what it concerns. A lot of Results are Statistics of Illumina sequencing data, which are not needed in this type of work. It is not known what specific bacteria affects Figure 2. Where are raw data of sequencing? Original sequencing files (fasta, fastq) should be provided to one of Databases (e.g. NCBI) and available to other scientists. What is reference number to the database? Where are information about other bacteria obtained in sequencing? Conclusions are strange "These results may provide clues for understanding at the intestinal microbiota level the mechanism of intestinal inflammation and, furthermore, the crispiness of grass carp fed with faba beans". Authors not studied inflammatory factors, and not studied crispiness, therefore do the conclusions relate to this publication for sure? Unfortunately, due to the serious microbiological errors of this work, lack of full raw data, conclusions not related to the article.

Reviewer 2 Report

Dear editor:

I appreciate the Editor to give me a chance to review this paper. This manuscript reported the effect of the effect of faba beans on the intestinal microbiota of grass carp. However, there are some issues in this paper.

Other data of grass carp, such as body weights, could be presented in this work. If the authors want to focus on the intestinal microbiota, maybe the full-length 16S rRNA gene for SMRT sequencing or metagenomics could be carried out. Because the gut microbiota in genus and species can be identified from sequencing of 16S rDNA V3-V4 region. I do not think that Acinetobacter dominates the intestinal microbiota of grass carp fed with faba beans, thus the title should be improved. “Statistics of effective/taxon tags and OTUs showed that there were significant differences (P-value <0.05) in the corresponding groups (TF/CF, TM/CM, TH/CH and T/C).”, “On the other hand, the numbers of OTUs from the hind-intestine were significantly less than those from the fore- and mid-intestine…” however, there was no statistical analysis in Table1.

Reviewer 3 Report

This study is an analysis of the microbial composition of grass carp fed with either a conventional diet or faba beans, which are known to alter the texture of the fish, presumably through inflammation of the muscliculture. The authors report that the faba bean diet shifts the microbial composition dramatically, reducing the proportion of Fusobacteria and increasing the proportion of Acinetobacter while increasing diversity overall. 

The data is presented clearly here and the statistical methodology is sound but the relationship between the phenomena described here and the conclusions suggested by the authors is unclear. In particular, the relationship between intestinal inflammation and muscular inflammation is ambiguous and there is no obvious way (including Tax4fun) to infer gut inflammation from changes in microbial composition. To attempt to bridge this gap between the one experiment described in this paper and the authors' goal of understanding this physiological shift, I would encourage the authors to examine not only inflammatory biomarkers but also shifts in gut aerobicity; fusobacteria are anaerobic while acinetobacter are commonly described as obligate aerobes. Despite the overall statistical soundness, I am also curious about two of the TH samples which vary significantly from the others, one enriched in Vibrio and another in Brevinema. This might argue that the sample size is insufficient to capture some of the trends here.